# GLASU: A Communication-Efficient Algorithm for Federated Learning with Vertically Distributed Graph Data

## Abstract

Vertical federated learning (VFL) is a distributed learning paradigm, where computing clients collectively train a model based on the partial features of the same set of samples they possess. Current research on VFL focuses on the case when samples are independent, but it rarely addresses an emerging scenario when samples are interrelated through a graph. In this work, we train a graph neural network (GNN) through VFL, where each client owns a part of the node features and a different edge set. This data scenario incurs a significant communication overhead, not only because of the handling of distributed features but also due to neighborhood aggregation in a GNN. Moreover, the training analysis is faced with a challenge caused by the biased stochastic gradients. We propose a model-splitting method that splits a backbone GNN across the clients and the server and a communication-efficient algorithm, GLASU, to train such a model. GLASU adopts lazy aggregation and stale updates to skip communication in neighborhood aggregation and in model updates, respectively, greatly reducing communication while enjoying convergence guarantees. We conduct extensive numerical experiments on real-world datasets, showing that GLASU effectively trains a GNN that matches the accuracy of centralized training, while using only a fraction of the time due to communication saving.

## 1 Introduction

Vertical federated learning (VFL) is a newly developed machine learning scenario in distributed optimization, where clients share data with the same sample identity but each client possesses only a subset of the features for each sample. The goal is for the clients to collaboratively learn a model based on all features. Such a scenario appears in many applications, including healthcare, finance, and recommendation systems.

Most of the current VFL solutions (Chen et al., 2020b; Liu et al., 2022) treat the case where samples are independent, but omit their relational structure. However, the pairwise relationship between samples emerges in many occasions and it can be crucial in several learning scenarios, including the low-labeling-rate scenario in semi-supervised learning and the no-labeling scenario in self-supervised learning.

Consider, for example, a company that offers news recommendations to its subscribed users. Several departments may be maintaining a separate user graph in their own compute infrastructure: a professional network where users are connected through occupational ties; a personal network where users are connected through personal life interactions; a follower network where a user is a follower of another on social media, etc. Further, the user data in each graph may contain different features (e.g., occupation related, life related, and interest related, respectively). To offer personal recommendations, the company sets up a server that communicates with each client (each department's computer), to train a model that predicts multiple labels for each user without revealing each client's raw local data. See Figure 1 for an illustration.

One of the most effective machine learning models for such a prediction task is graph neural networks (GNNs) (Kipf & Welling, 2016; Hamilton et al., 2017; Chen et al., 2018; Velickovic et al., 2018; Chen et al., 2020a). This model performs neighborhood aggregation in every feature transformation layer, such that the prediction of a graph node is based on not only the information of this node but also that of its neighbors.

VFL on graph-structured data is not as well studied as that on other data, in part be-

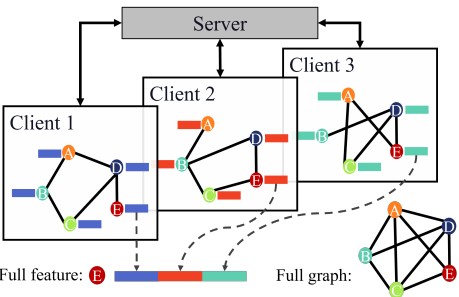

Figure 1: Data isolation of vertically distributed graph-structured data over three clients.

cause of the challenges incurred by an enormous amount of communication. The communication overhead comes not only from the aggregation of the partial features/representations of a datum, but also from the neighborhood aggregation unique to GNNs. That is, communication occurs in each layer of the neural network, so that the latest representation of a neighboring node can be used to update the representation of the center node. One solution to reduce communication is that each client uses a local GNN to extract node representations from its own graph and the server aggregates these representations to make predictions (Zhou et al., 2020). The drawback of this method is that the partial features of a node outside one client's neighborhood are not used, even if this node appears in another client's neighborhood. Another solution is to simulate centralized training: intermediate representations of each node are aggregated by the server, from where neighborhood aggregation is performed (Ni et al., 2021). This method suffers the communication overhead incurred in each layer computation.

In this work, we propose GLASU for communication-efficient VFL on graph data. The GNN model is split across the clients and the server, such that the clients can use a majority of existing GNNs as the backbone, while the server contains no model parameters. The server only aggregates and disseminates processed data (e.g., node embeddings) with the clients. The communication frequency between the clients and the server is mitigated through *lazy aggregation and stale updates* (hence the name of the method). For an $L$-layer GNN, GLASU communicates partial node representations only in $K$ layers and in every other $Q$ iterations, enjoying the reduction of communication by a factor of $QL/K$. GLASU can be considered as a framework that encompasses several well-known models and algorithms as special cases, including Liu et al. (2022) when the graphs are absent, Zhou et al. (2020) when all aggregations but the final one are skipped ($K = 1$), Ni et al. (2021) when no aggregations are skipped ($K = L$), and centralized training when only a single client exists.

With the enjoyable reduction in communication, another difficulty is the convergence analysis, which admits two challenges: the biased gradient caused by neighborhood sampling in training GNNs and the correlated updates due to the use of stale node representations. We conduct an analysis based on the error decomposition of the gradient, showing that the training admits a convergence rate of $\mathcal{O}((TQ)^{-1})$, where $T$ is the number of training rounds, each of which contains $Q$ iterations.

We summarize the main contributions of this work below:

1. Model design: We propose a flexible, federated GNN architecture that is compatible with a majority of existing GNN backbones.
2. Algorithm design: We propose the communication-efficient GLASU algorithm to train the model. Therein, lazy aggregation saves communication for each joint inference round, through skipping some aggregation layers in the GNN; while stale updates further save communication by allowing the clients to use stale global information for multiple local model updates.
3. Theoretical analysis: We provide theoretical convergence analysis for GLASU by addressing the challenges of biased stochastic gradient estimation caused by neighborhood sampling and correlated update steps caused by using stale global information. To the

best of our knowledge, this is the first convergence analysis for federated learning with graph data.

4. Numerical results: We conduct extensive experiments on seven datasets, together with ablation studies, to demonstrate that GLASU can achieve a comparable performance as the centralized model on multiple datasets and multiple GNN backbones, and that GLASU effectively saves communication and reduces training time.

## 2 Problem, background, and related works

**Problem setup:** Consider $M$ clients, indexed by $m = 1, \ldots, M$, each of which holds a part of a graph with the node feature matrix $\mathbf{X} \in \mathbb{R}^{N \times d}$ and the edge set $\mathcal{E}$. Here, $N$ is the number of nodes in the graph and $d$ is the feature dimension. The number of clients is restricted by the feature dimension and is typically small. We assume that each client has the same node set and the same set of training labels, $\mathbf{y}$, but a different edge set $\mathcal{E}_m$ and a non-overlapping node feature matrix $\mathbf{X}_m \in \mathbb{R}^{N \times d_m}$, such that $\mathcal{E} = \bigcup_{m=1}^{M} \mathcal{E}_m$, $\mathbf{X} = [\mathbf{X}_1, \ldots, \mathbf{X}_M]$, and $d = \sum_{m=1}^{M} d_m$. We denote the client dataset as $\mathcal{D}_m = \{\mathbf{X}_m, \mathcal{E}_m, \mathbf{y}\}$ and the full dataset as $\mathcal{D} = \{\mathbf{X}, \mathcal{E}, \mathbf{y}\}$. The task is for the clients to collaboratively infer the labels of nodes in the test set.

### 2.1 Graph convolutional network

The graph convolution network (GCN) (Kipf & Welling, 2016) is a typical example of the family of GNNs. Inside GCN, a graph convolution layer reads

$$\mathbf{H}[l+1] = \sigma\Big(\mathbf{A}(\mathcal{E}) \cdot \mathbf{H}[l] \cdot \mathbf{W}[l]\Big), \tag{1}$$

where $\sigma(\cdot)$ denotes the point-wise nonlinear activation function, $\mathbf{A}(\mathcal{E}) \in \mathbb{R}^{N \times N}$ denotes the adjacency matrix defined by the edge set $\mathcal{E}$ with proper normalization, $\mathbf{H}[l] \in \mathbb{R}^{N \times d[l]}$ denotes the node representation matrix at layer $l$, and $\mathbf{W}[l] \in \mathbb{R}^{d[l] \times d[l+1]}$ denotes the weight matrix at the same layer. The initial node representation matrix $\mathbf{H}[0] = \mathbf{X}$. The classifier is denoted as $\hat{\mathbf{y}} = f(\mathbf{H}[L], \mathbf{W}[L])$ with weight matrix $\mathbf{W}[L]$ and the loss function is denoted as $\ell(\mathbf{y}, \hat{\mathbf{y}})$. Therefore, the overall model parameter is $\mathbf{W} = \{\mathbf{W}[0], \ldots, \mathbf{W}[L-1], \mathbf{W}[L]\}$.

Mini-batch training of GCN (and GNNs in general) faces a scalability challenge, because computing one or a few rows of $\mathbf{H}[L]$ (i.e., the representations of a mini-batch) requires more and more rows of $\mathbf{H}[L-1]$, $\mathbf{H}[L-2]$, ... recursively, in light of the multiplication with $\mathbf{A}(\mathcal{E})$ in (1). This is known as the *explosive neighborhood problem* unique to graph-structured data. Several sampling strategies were proposed in the past to mitigate the explosion; in this work, we adopt the layer-wise sampling proposed by FastGCN (Chen et al., 2018). Starting from the output layer $L$, which is associated with a mini-batch of training nodes, $\mathcal{S}[L]$, we iterate over the layers backward such that at layer $l$, we sample a subset of neighbors for $\mathcal{S}[l+1]$, namely $\mathcal{S}[l]$. In doing so, at each layer, we form a bipartite graph with edge set $\mathcal{E}[l] = \{(i,j)|i \in \mathcal{S}[l+1], j \in \mathcal{S}[l]\}$. Then, each graph convolution layer becomes

$$\mathbf{H}[l+1][\mathcal{S}[l+1]] = \sigma\Big(\mathbf{A}(\mathcal{E}[l]) \cdot \mathbf{H}[l][\mathcal{S}[l]] \cdot \mathbf{W}[l]\Big), \tag{2}$$

where $\mathbf{A}(\mathcal{E}[l]) \in \mathbb{R}^{|\mathcal{S}[l+1]| \times |\mathcal{S}[l]|}$ is a properly scaled submatrix of $\mathbf{A}(\mathcal{E})$ and $\mathbf{H}[l][\mathcal{S}[l]]$ denotes the rows of $\mathbf{H}[l]$ corresponding to the sampled neighbor set $\mathcal{S}[l]$.

### 2.2 Related works

**Vertical federated learning** is a learning paradigm where the features of the data are distributed across clients, who collaborate to train a model that incorporate all features (Liu et al., 2022; Chen et al., 2020b; Romanini et al., 2021; Yang et al., 2019b; Gu et al., 2021; Yang et al., 2019a; Xu et al., 2021). Thus, the global model is split among clients and the key challenge is the heavy communication costs on exchanging partial sample information for computing the losses and the gradients for each sample. Most works consider simple models (e.g., linear) because complex models incur multiple rounds of communication for prediction.

**Federated learning with graphs** includes four scenarios. The *graph-level* scenario is *horizontal*, where each client possesses a collection of graphs and all clients collaborate to

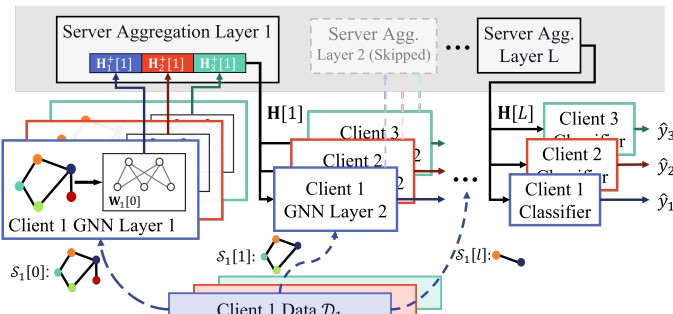

Figure 2: Illustration of the split model on $M = 3$ clients with lazy aggregation. In the model, the second server aggregation layer is skipped and the graph size used by each layer gradually decreases, due to neighborhood aggregation (inverse of neighborhood sampling).

train a unified model (Zhang et al., 2021a; He et al., 2021; Bayram & Rekik, 2021; Xie et al., 2021). The task is to predict graph properties (such as molecular properties).

The *subgraph-level* scenario could be either *vertical* or *horizontal*. In the vertical scenario, each client holds a part of the node features, a part of the whole model, and additionally a subgraph of the global graph (Zhou et al., 2020; Ni et al., 2021). The clients aim to collaboratively train a model (combined from those of each client) to predict node properties (such as the category of a paper in a citation network). Our work addresses this scenario.

The *subgraph-level, horizontal* scenario, on the other hand, considers training a GNN for node property prediction in a *distributed* manner: a graph is partitioned and each client holds one partition (Zhang et al., 2021b; Wu et al., 2021; Chen et al., 2022; Yao & Joe-Wong, 2022). A challenge to address is the aggregation of information along edges crossing different clients. This scenario differs from the vertical scenario in that features are not partitioned among clients and the graph partitions do not overlap.

The fourth scenario is *node-level*: the clients are connected by a graph and thus each of them is treated as a node. In other words, the clients, rather than the data, are graph-structured. It is akin to *decentralized learning*, where clients communicate to each other via the graph to train a unified model (Lalitha et al., 2019; Meng et al., 2021; Caldarola et al., 2021; Rizk & Sayed, 2021).

Due to the space limitation, please see Appendix A for in-depth discussions of the related works.

## 3 PROPOSED APPROACH

In this section, we present the proposed model and the training algorithm GLASU for federated learning on vertically distributed graph data. The neighborhood aggregation in GNNs poses communication challenges distinct from conventional VFL. To mitigate this challenge, we propose lazy aggregation and stale updates to effectively reduce the communication between the clients and the server, while maintaining comparable prediction performance as centralized models. For notational simplicity, we present the approach by using the full-graph notation (1) but note that the implementation involves neighborhood sampling, where a more precise notation should follow (2), and that one can easily change the backbone from GCN to other GNNs.

### 3.1 GNN MODEL SPLITTING

We split the GNN model among the clients and the server, approximating a centralized model. Specifically, each GNN layer contains two sub-layers: the client GNN sub-layer and the server aggregation sub-layer. At the $l$-th layer, each client computes the local feature matrix

$$\mathbf{H}_m^+[l] = \sigma\Big(\mathbf{A}(\mathcal{E}_m) \cdot \mathbf{H}_m[l] \cdot \mathbf{W}_m[l]\Big)$$

with the local weight matrix $\mathbf{W}_m[l]$ and the local graph $\mathcal{E}_m$, where we use the superscript $^+$ to denote local representations before aggregation. Then, the server aggregates the clients' representations and outputs $\mathbf{H}[l+1]$ as

$$\mathbf{H}[l+1] = \text{Agg}(\mathbf{H}_1^+[l], \ldots, \mathbf{H}_M^+[l]),$$

where $\mathrm{Agg}(\cdot)$ is an aggregation function. In this paper, we only consider parameter-free aggregations, including averaging and concatenation. The server broadcasts the aggregated $\mathbf{H}[l+1]$ to the clients so that computation proceeds to the next layer. In the final layer, each client computes a prediction. This layer is the same among clients because they receive the same $\mathbf{H}[L]$.

The two aggregation operations of our choice render a rather simple implementation of the server. They bring in two advantages: parameter-free and memory-less. Since the operations do not contain any learnable parameters, the server does not need to perform gradient computations. Moreover, in the backward pass, these operations do not require data from the forward pass to back-propagate the gradients (memory-less). Specifically, for averaging, the server back-propagates $\frac{1}{M}\nabla_{\mathbf{H}[l+1]}\mathcal{L}$ to each client, where $\mathcal{L}$ denotes the loss; while for concatenation, the server back-propagates the corresponding block of $\nabla_{\mathbf{H}[l+1]}\mathcal{L}$.

We illustrate in Figure 2 the split of each GNN layer among the clients and the server. Note the difference of our approach from existing approaches. Our model splitting resembles federated split learning (SplitFed) (Thapa et al., 2022); but in SplitFed, each client can collaborate with the server to perform inference or model updates without accessing information from other clients, whereas in our case, all clients collectively perform the job. Our approach also differs from conventional VFL that splits the local feature processing and the final classifier among the clients and the server respectively, such that each model update requires a single U-shape communication (Chen et al., 2020b). In our case, due to the graph structure, each GNN layer contains one client-server interaction and the number of interactions is equal to the number of GNN layers (we will relax this in the following subsection).

## 3.2 LAZY AGGREGATION

The development in the preceding subsection approximates a centralized model, but it is not communication friendly because each layer requires one round of client-server communication. We propose two communication-saving strategies in this subsection and the next. We first consider *lazy aggregation*, which skips aggregation in certain layers.

Instead of performing server aggregation at each layer, we specify a subset of $K$ indices, $\mathcal{I} = \{l_1, \ldots, l_K\} \subset [L]$, such that aggregation is performed only at these layers. That is, at a layer $l \in \mathcal{I}$, the server performs aggregation and broadcasts the aggregated representations to the clients, serving as the input to the next layer: $\mathbf{H}_m[l+1] = \mathbf{H}[l+1]$; while at a layer $l \notin \mathcal{I}$, each client uses the local representations as the input to the next layer: $\mathbf{H}_m[l+1] = \mathbf{H}_m^+[l]$. By doing so, the amount of communication is reduced from $\mathcal{O}(L)$ to $\mathcal{O}(K)$.

There are subtleties caused by neighborhood sampling, similar to those faced by FastGCN (see Section 2.1). First, it requires additional rounds of communication to synchronize the sample indices, because whenever server aggregation is performed, it must be done on the same set of sampled nodes across clients. Hence, in the additional communication rounds, the server takes the union of the clients' index sets $\mathcal{S}_m[l_k]$ and broadcasts $\mathcal{S}[l_k] = \bigcup_{m=1}^M \mathcal{S}_m[l_k]$ to the clients. Second, when server aggregation is skipped at a layer $l \notin \mathcal{I}$, each client can use its own set of sampled nodes, $\mathcal{S}_m[l]$, which may differ from each other. Such a procedure is more flexible than conventional VFL where sample features are generally processed synchronously. The sampling procedure is summarized in Algorithm 2 in Appendix B.1.

## 3.3 STALE UPDATES

To further reduce communication, we consider *stale updates*, which skip aggregation in certain iterations and use stale node representations to perform model updates. The key idea is to use the same mini-batch, including the sampled neighbors at each layer, for training $Q$ iterations. In every other $Q$ iterations, the clients store the aggregated representations at the server aggregation layers. Then, in the subsequent iterations, every server aggregation is replaced by a local aggregation between a client's up-to-date node representations and other clients' stale node representations. By doing so, the clients and the server only need to communicate once in every $Q$ iterations.

---

**Algorithm 1** Training Procedure. All referenced algorithms are detailed in Appendix B.1.

> **for** $t = 0, \ldots, T$ **do**
>> **Server/Client** (Algorithm 2): Sample $\{\mathcal{S}_m^t[l]\}_{l=0}^L$.
>>
>> **Client**: $\mathbf{W}_m^{t,0} = \begin{cases} \mathbf{W}_m^{t-1,Q}, & t > 0 \\ \mathbf{W}_m^0, & t = 0 \end{cases}$.
>>
>> **Server/Client** (Algorithm 3): $\{\mathbf{H}_{-m}^t[l+1]\}_{l \in \mathcal{I}} = \mathbf{JointInference}(\mathbf{W}_m^{t,0}, \mathcal{D}_m, \{\mathcal{S}_m^t[l]\}_{l=0}^L)$.
>> **for** $q = 0, \ldots, Q-1$ **do**
>>> **Client** (Algorithm 4): $\mathbf{W}_m^{t,q+1} = \mathbf{LocalUpdate}(\mathbf{W}_m^{t,q}, \mathcal{D}_m, \{\mathcal{S}_m^t[l]\}_{l=0}^L, \{\mathbf{H}_{-m}^t[l+1]\}_{l \in \mathcal{I}})$.
>> **end for**
> **end for**
> **Output:** $\{\mathbf{W}_m^{T,Q}\}_{m=1}^M$

---

Specifically, let a round of training contain $Q$ iterations and use $t$ to index the rounds. At the beginning of each round, the clients and the server jointly decide the set of nodes used for training at each layer. Then, they perform a joint inference on the representations $\mathbf{H}_m^{t,+}[l]$ at every layer $l \in \mathcal{I}$. Each client $m$ will store the "all but $m$" representation $\mathbf{H}_{-m}^t[l+1]$ through extracting such information from the aggregated representations $\mathbf{H}_m^t[l+1]$:

$$\mathbf{H}_{-m}^t[l+1] = \text{Extract}(\mathbf{H}_m^t[l+1], \mathbf{H}_m^{t,+}[l]).$$

For example, when the server aggregation is averaging, the extraction is

$$\text{Extract}(\mathbf{H}_m^t[l+1], \mathbf{H}_m^{t,+}[l]) = \mathbf{H}_m^t[l+1] - \frac{1}{M}\mathbf{H}_m^{t,+}[l],$$

Afterward, the clients perform $Q$ iterations of model updates, indexed by $q = 0, \ldots, Q-1$, on the local parameters $\mathbf{W}_m^{t,q}$ in parallel, using the stored aggregated information $\mathbf{H}_{-m}^t[l+1]$ to perform local computation, replacing server aggregation. The name "stale updates" comes from the fact that $\mathbf{H}_{-m}^t[l+1]$ is computed by using stale model parameters $\{\mathbf{W}_{m'}^{t,0}\}_{m' \neq m}$ at all iterations $q \neq 0$. The extraction and the local updates are summarized in Algorithm 3 and Algorithm 4, respectively, in Appendix B.1.

### 3.4 SUMMARY

The overall training procedure is summarized in Algorithm 1. For communication savings, lazy aggregation brings in a factor of $L/K$ and stale updates bring in a factor of $Q$. Therefore, the overall saving factor is $QL/K$. Note that the algorithm assumes that all clients have the training labels. If the labels can be held by only one client (say, A), a slight modification by broadcasting the gradient with respect to the final-layer output possessed by A, suffices. See Appendix B.2 for details. Although privacy is not the major focus of this paper, we argue that GLASU is also compatible with existing privacy-preserving approaches and provide discussion in Appendix C.

### 3.5 SPECIAL CASES

It is interesting to note that GLASU encompasses several well-known methods as special cases.

**Conventional VFL.** VFL algorithms can be viewed as a special case of GLASU, where $\mathbf{A}(\mathcal{E}_m) = \mathbf{I}$ for all $m$. In this case, no neighborhood sampling is needed and GLASU reduces to Liu et al. (2022).

**Existing VFL algorithms for graphs.** The model of Zhou et al. (2020) is a special case of GLASU, with $K = 1$; i.e., no communication is performed except the final prediction layer. In this case, the clients omit the connections absent in the self subgraph but present in other clients' subgraphs. The model of Ni et al. (2021) is also a special case of GLASU, with $K = L$. This case requires communication at all layers and is less efficient.

**Centralized GNNs.** When there is a single client ($M = 1$), our setting is the same as centralized GNN training. Specifically, by letting $K = L$ and properly choosing the server aggregation function $\text{Agg}(\cdot)$, our split model can achieve the same performance as a centralized GNN model. Note that using lazy aggregation ($K \neq L$) and choosing the server aggregation function as concatenation or averaging will make the split model different from a centralized GNN.

## 4 CONVERGENCE ANALYSIS

In this section, we analyze the convergence behavior of GLASU under lazy aggregation and stale updates. To start the analysis, denote by $\mathcal{S}^t = \{\mathcal{S}^t_m[l]\}^{L,M}_{l=1,m=1}$ the samples used at round $t$ (which include all sampled nodes at different layers and clients); by $S = |\mathcal{S}^t_m[L]|$ the batch size; and by $\mathcal{L}(\mathbf{W}; \mathcal{S})$ the training objective, which is evaluated at the overall set of model parameters across clients, $\mathbf{W} = \{\mathbf{W}_m\}^M_{m=1}$, and a batch of samples, $\mathcal{S}$.

A few assumptions are needed (see Appendix D.1 for formal statements). **A1**: The loss function $\ell$ is $G_\ell$-smooth with $L_\ell$-Lipschitz gradient; and a client's prediction function $f_m$ is $G_f$-smooth with $L_f$-Lipschitz gradient. **A2**: The training objective $\mathcal{L}(\mathbf{W}; \mathcal{D})$ is bounded below by a finite constant $\mathcal{L}^\star$. **A3**: The samples $\mathcal{S}^t$ are uniformly sampled from the neighbor set in each layer.

**Theorem 1.** *Under assumptions A1–A3, by running Algorithm 1 with constant step size $\eta \le C_0^{-1} \cdot (1 + 2Q^2 M)^{-1}$, with probability at least $p = 1 - \delta$, the averaged squared gradient norm is bounded by:*

$$\frac{1}{TQ} \sum_{t=0}^{T-1} \sum_{q=0}^{Q-1} \mathbb{E} \left\| \nabla \mathcal{L}(\mathbf{W}^{t,q}; \mathcal{D}) \right\|^2 \le \frac{2\Delta_{\mathcal{L}}}{\eta TQ} + \frac{28 \eta M \cdot (C_0 + \sqrt{M+1}Q)}{3} \sigma,$$

*where* $\Delta_{\mathcal{L}} = \mathcal{L}(\mathbf{W}^{0,0}) - \mathcal{L}^\star$, $C_0 = G_\ell L_f + L_\ell G_f^2$, *and* $\sigma > 0$ *is a function of* $\log(TQ/\delta), L_f, L_g, G_f$ *and* $G_g$.

*Remark* 1. There are two key challenges in the analysis. (1) Owing to neighborhood sampling, the stochastic gradient is biased (i.e., $\mathbb{E}_{\mathcal{S}} \nabla \mathcal{L}(\mathbf{W}; \mathcal{S}) \ne \nabla \mathcal{L}(\mathbf{W}; \mathcal{D})$). (2) The stale updates in one communication round are correlated, as they use the same mini-batch and samples. Hence, the general unbiasedness and independence assumptions on the stochastic gradients in the analysis of SGD-type of algorithms do not apply. We borrow the technique by Ramezani et al. (2020) to bound the error of the stochastic gradient through the bias-variance decomposition and extend the analysis by Liu et al. (2022) for VFL with correlated updates to establish our proof. For details, see Appendix D.

*Remark* 2. To better expose the convergence rate, assuming that $Q$ is upper bounded by $\frac{C_0}{\sqrt{M+1}}$, one may set $\eta = \sqrt{\frac{3\Delta_{\mathcal{L}}}{28MC_0\sigma TQ}}$, such that

$$\frac{1}{TQ} \sum_{t=0}^{T-1} \sum_{q=0}^{Q-1} \mathbb{E} \left\| \nabla \mathcal{L}(\mathbf{W}^{t,q}; \mathcal{D}) \right\|^2 \le 8 \sqrt{\frac{7\Delta_{\mathcal{L}} M C_0 \sigma}{3TQ}}.$$

Ignoring the logarithmic factor $\log(TQ/\delta)$ in $\sigma$, the above bound states that the squared gradient norm decreases as $\mathcal{O}((TQ)^{-1})$. Note that this bound holds only when $T$ is sufficiently large, because the choice of $\eta$ must satisfy the condition of Theorem 1.

*Remark* 3. Based on the preceding remark, we see that to achieve $\epsilon$-stationarity, the number of model updates is $QT = \mathcal{O}(\frac{1}{\epsilon^2})$. That is, as long as $Q$ obeys the upper bound, running more local updates ($Q$) reduces the amount of communications ($T$). To the best of our knowledge, this is the first result for VFL on graph data.

*Remark* 4. While we have analyzed the impact of stale updates ($Q$), lazy aggregation ($K$) does not play a role in convergence, because it does not affect model updates. Instead, it affects model accuracy in a manner similar to how changing a neural network impacts the prediction accuracy.

*Remark* 5. If we consider the impact of the number of clients, the factor $M$ in the numerator of the bound indicates a slowdown when more clients participate training. Similar results are seen in FedBCD (Liu et al., 2022), but therein one can use a large batch size $S$ to counter the slowdown. For graphs, however, $S$ does not appear in the bound because of the biased gradient estimation. Nevertheless, we note that unlike other federated scenarios, in VFL, $M$ is very small because it is limited by, e.g., the feature length.

## 5 NUMERICAL EXPERIMENTS

In this section, we conduct numerical experiments on a variety of datasets and demonstrate the effectiveness of GLASU in training with distributed graph data. We first compare

Table 1: Test accuracy (%). The compared algorithms are Centralized training (Cent.); Standalone training (StAl.); Simulated centralized training (Sim.); GLASU with no stale updates, i.e., $Q = 1$ (GLASU-1); and GLASU with stale updates $Q = 4$ (GLASU-4).

| Dataset | Cent. | StAl. | Sim. | GLASU-1 | GLASU-4 |
|---|---|---|---|---|---|
| Cora | $80.9 \pm 0.6$ | $74.6 \pm 0.5$ | $80.1 \pm 1.2$ | $81.0 \pm 1.3$ | $80.3 \pm 1.2$ |
| PubMed | $84.9 \pm 0.6$ | $77.2 \pm 0.5$ | $82.7 \pm 1.2$ | $82.3 \pm 1.6$ | $83.8 \pm 1.8$ |
| CiteSeer | $70.2 \pm 0.8$ | $64.4 \pm 0.5$ | $70.0 \pm 1.2$ | $70.0 \pm 1.7$ | $68.8 \pm 3.3$ |
| Suzhou | $94.3 \pm 0.3$ | $51.6 \pm 0.9$ | $93.5 \pm 0.6$ | $92.7 \pm 1.4$ | $90.4 \pm 0.8$ |
| Venice | $95.7 \pm 0.5$ | $33.5 \pm 2.1$ | $93.1 \pm 1.3$ | $92.2 \pm 0.6$ | $91.0 \pm 1.6$ |
| Amsterdam | $94.6 \pm 0.1$ | $59.8 \pm 1.0$ | $95.5 \pm 0.8$ | $93.1 \pm 0.8$ | $94.9 \pm 0.4$ |
| Reddit | $95.6 \pm 0.1$ | $87.3 \pm 0.3$ | $95.3 \pm 0.7$ | $95.7 \pm 0.6$ | $94.7 \pm 1.1$ |

its performance with related methods, including those tackling a different assumption on the data distribution and communication pattern. Then, we examine the communication saving owing to the use of lazy aggregation and stale updates. We further showcase the flexibility of GLASU through demonstration with different GNN backbones and varying clients. The experiments are conducted on a distributed cluster with three Tesla V100 GPUs communicated through Ethernet.

## 5.1 DATASETS

We use seven datasets (in three groups) with varying sizes and data distributions: the Planetoid collection (Yang et al., 2016), the HeriGraph collection (Bai et al., 2022), and the Reddit dataset (Hamilton et al., 2017). Each dataset in the HeriGraph collection (Suzhou, Venice, and Amsterdam) contains data readily distributed: three subgraphs and more than three feature blocks for each node. Hence, we use three clients, each of which handles one subgraph and one feature block. For the other four datasets (Cora, PubMed, and CiteSeer in the Planetoid collection; and Reddit), each contains one single graph and thus we manually construct subgraphs through randomly sampling the edges and splitting the input features into non-overlapping blocks, so that each client handles one subgraph and one feature block. The dataset statistics are summarized in Table 4 and more details are given in Appendix E.1.

## 5.2 ACCURACY

We compare GLASU with three training methods: (a) centralized training, where there is only a single client ($M = 1$), which holds the whole dataset without any data distribution and communication; (b) standalone training (Zhou et al., 2020), where each client trains a model with its local data only and they do not communicate; (c) simulated centralized training (Ni et al., 2021), where each client possesses the full graph but only the partial features, so that it simulates centralized training through server aggregation in each GNN layer. Methods (b) and (c) are typical VFL methods; they are also special cases of our method (see Section 3.5). Except for centralized training, the number of clients $M = 3$. The number of training rounds, $T$, and the learning rate $\eta$ are optimized through grid search. See Appendix E.2 for details.

We use GCNII (Chen et al., 2020a) as the backbone GNN. GCNII improves over GCN through including two skip connections, one with the current layer input and the other with the initial layer input. We set the number of layers $L = 4$ and the mini-batch size $S = 16$. For neighborhood sampling, the sample size is three neighbors per node on average. We set $K = 2$; i.e., lazy aggregation is performed in the middle and the last layer.

Table 1 reports the average classification accuracy of GLASU and the compared training methods, repeated five times. As expected, standalone training produces the worst results, because each client uses only local information and misses edges and node features present in other clients. The centralized training and its simulated version lead to similar performance, also as expected, because server aggregation (or its equivalence in centralized training) on each GNN layer takes effect. Our method GLASU, which skips half of the aggregations, yields prediction accuracy rather comparable with these two methods. Using stale updates ($Q = 4$) is generally outperformed by no stale updates ($Q = 1$), but occasionally it is better (see PubMed and Amsterdam). The gain in using lazy aggregation and stale updates occurs in timing, as will be demonstrated next.

Table 2: Test accuracy (%), runtime (seconds), and saving in runtime (%) under different numbers of lazy aggregation layers ($K = 4, 2, 1$). The saving is with respect to $K = 4$. Left: PubMed; right: Amsterdam.

| # Layer | $K = 4$ | $K = 2$ | $K = 1$ | $K = 4$ | $K = 2$ | $K = 1$ |
|---|---|---|---|---|---|---|
| Accuracy | $82.5 \pm 1.0$ | $83.8 \pm 1.8$ | $82.2 \pm 0.7$ | $93.6 \pm 0.7$ | $94.9 \pm 0.4$ | $92.0 \pm 1.7$ |
| Runtime | $130 \pm 12$ | $96.6 \pm 9.9$ | $81.3 \pm 6.5$ | $913 \pm 76$ | $544 \pm 44$ | $382 \pm 35$ |
| Saving | $-$ | $25.7$ | $37.5$ | $-$ | $40.4$ | $58.2$ |

Table 3: Test accuracy (%) and runtime (seconds) under different numbers of stale updates ($Q = 2, 4, 6, 16$) for the same accuracy threshold. Top: PubMed (threshold: 82%); bottom: Amsterdam (threshold: 89%).

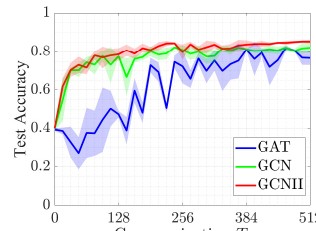

Figure 3: Test accuracy under three backbone GNNs on PubMed.

| # Stale | $Q = 2$ | $Q = 4$ | $Q = 8$ | $Q = 16$ |
|---|---|---|---|---|
| Accuracy | $82.5 \pm 1.6$ | $82.0 \pm 2.4$ | $82.1 \pm 0.3$ | N/A |
| Runtime | $66.1 \pm 5.0$ | $43.8 \pm 4.0$ | $88.9 \pm 7.4$ | $> 128$ |

| # Stale | $Q = 2$ | $Q = 4$ | $Q = 8$ | $Q = 16$ |
|---|---|---|---|---|
| Accuracy | $89.2 \pm 0.4$ | $89.3 \pm 0.7$ | $90.7 \pm 0.5$ | $90.3 \pm 1.1$ |
| Runtime | $1323 \pm 44$ | $521 \pm 44$ | $324 \pm 31$ | $250 \pm 24$ |

### 5.3 COMMUNICATION SAVING

To further investigate how the two proposed techniques affect the model performance and save the communication, we conduct a study on (a) the lazy aggregation parameter $K$ and (b) the stale update parameter $Q$.

**Lazy aggregation:** We use a 4-layer GCNII as the backbone and set $K = 1, 2, 4$. The aggregation layers are "uniform" across the model layers. That is, when $K = 1$, server aggregation is performed on the last layer; when $K = 2$, on the middle layer and the last layer; and when $K = 4$, on all layers. The test accuracy and runtime are listed in Table 2. We observe that the runtime decreases drastically when using fewer and fewer aggregation layers: from $K = 4$ to $K = 1$, the reduction is 37.5% for PubMed and 58.2% for Amsterdam. The accuracy is comparable in all cases.

**Stale updates:** We experiment with a few choices of $Q$: 2, 4, 8, and 16. We report the time to reach the same test accuracy threshold in Table 3. We see that stale updates help speed up training by using fewer communication rounds, corroborating Remark 3 of the theory in Section 4. This trend occurs on the Amsterdam dataset even when taking $Q$ as large as 16. The trend is also noticeable on PubMed, but at some point ($Q = 8$) it is reverted, likely because it gets harder and harder to reach the accuracy threshold. We speculate that the target 82% can never be achieved at $Q = 16$. This observation is consistent with Remark 2 of the theory, requiring $Q$ to be upper bounded to claim $\mathcal{O}((TQ)^{-1})$ convergence.

### 5.4 FLEXIBILITY

To demonstrate the flexibility of GLASU, we conduct experiments to show the performance under (a) different GNN backbones and (b) different numbers of clients, $M$. We first test GLASU on three representative GNN backbones: GCN, GAT (Velickovic et al., 2018), and GCNII. The test accuracy over training rounds is plotted in Figure 3. We set $M = 3, 5, 7$ and investigate the change in performance for different training methods. Due to space limitations, we include the experiment results and the discussions in Appendix E.3.

## 6 CONCLUSION

We have presented a flexible model splitting approach for VFL with vertically distributed graph data and proposed a communication-efficient algorithm, GLASU, to train the resulting GNN. Due to the graph structure, VFL on GNNs incurs heavy communication and poses an extra challenge in the convergence analysis, as the stochastic gradients are no longer unbiased. To overcome these challenges, our approach uses lazy aggregation to skip server-client communication and stale global information to update local models, leading to significant communication reduction. Our analysis makes no assumptions on unbiased gradients. We provide extensive experiments to show the flexibility of the model and the communication saving in training, without compromise on the model quality.

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
