# OpenReview forum: "GLASU: A Communication-Efficient Algorithm for Federated Learning with Vertically Distributed Graph Data"
_ICLR.cc/2024/Conference — Submitted to ICLR 2024_

### Official Review · Reviewer_zunS · 2023-10-13

**Soundness:** 2 fair
**Presentation:** 3 good
**Contribution:** 2 fair
**Rating:** 5
**Confidence:** 3

**Summary:**

This paper proposes GLASU, a method for vertical federated GNN. The key innovations of this paper are two techniques, lazy aggregation (i.e. not all GNN layers require forward and backward aggregation), and stale updating (i.e. clients store a stale version of neighbor embeddings which can skip some cross-client embedding communication). The authors provide convergence analysis about their proposed GLASU. Numerical experiments also show the effectiveness of the proposed GLASU over standalone baseliens.

**Strengths:**

1. The studied problem is of practical value. VFL and federated graph learning are both practical problems that are useful in reality. In addition, the fact that GNNs require message passing between nodes actually worsens the communication overhead of VFL. Thus, the two problems indeed have something in common, i.e. communication overhead, so combining these two problems and addressing them in one work makes sense.

2. The paper is generally well written, well organized and easy to understand.

**Weaknesses:**

1. The technical solutions to the problems seem pretty straightforward and are somehow not new. The stale updating method specifically has been proposed in VFL (albeit not GNN) by (Fu et al. 2022) and (Liu et al. 2022). A method similar to the lazy aggregation method (avoid some neighborhood aggregation) is proposed in (Peng et al. 2022). Thus, although the paper indeed address an important combination of problems, the techniques are not new.

2. The problem setting seems narrow. Throughout the paper the authors seem to assume that all clients hold an identical set of nodes, and the algorithm can only operate on them, which may seem too restrictive. A set of fully aligned nodes is uncommon for different clients in VFL, and imposing such a constraint makes the setting narrow. At least, the authors should consider how to deal with mis-aligned nodes at individual clients. Also, the assumption that all clients have access to (the same set of) labels is also very strong. It is more likely that some may not have labels, or that they have distinct label spaces.

3. The experiments seem to have been set at a very simple level. First, the datasets used in this paper are simple (especially, Cora and Citeseer), as they have a limited number of nodes. Second, the 3-client setting seems small. Third, it is not clear why Sim. (i.e. simulating a centralized GNN) will lead to accuracy loss, and GLASU-1 can lead to accuracy improvements. Finally, it is also unclear why increased staleness leads to performance improvements in Table 3 (Amsterdam).

(Peng et al. 2022) Sancus: Staleness-Aware Communication-Avoiding Full-Graph Decentralized Training in Large-Scale Graph Neural Networks. VLDB 2022
(Liu et al. 2022) FedBCD: A Communication-Efficient Collaborative Learning Framework for Distributed Features. IEEE Trans on Signal Processing 2022
(Fu et al. 2022) Towards Communication-efficient Vertical Federated Learning Training via Cache-enabled Local Updates. VLDB 2022

**Questions:**

Maybe the authors want to clarify Weaknesses 1, 2, and 3.

---

### Official Review · Reviewer_fTEG · 2023-10-29

**Soundness:** 2 fair
**Presentation:** 3 good
**Contribution:** 2 fair
**Rating:** 3
**Confidence:** 4

**Summary:**

The paper proposes GLASU, a communication-efficient approach for vertical federated learning (VFL) on graph data using graph neural networks (GNNs). GLASU addresses the challenge of communication overhead in VFL by splitting the GNN model across clients and the server and employing lazy aggregation and stale updates. The authors conduct extensive experiments and provide theoretical convergence analysis, demonstrating the effectiveness of GLASU in achieving comparable performance to centralized models while reducing communication and training time.

**Strengths:**

1. The paper introduces a flexible federated GNN architecture that is compatible with existing GNN backbones, allowing for versatility in the choice of models.

2. The paper presents a communication-efficient approach by utilizing lazy aggregation and stale updates for Federated Learning with vertically distributed graph data.

3. The paper provides a theoretical convergence analysis for GLASU, addressing challenges related to biased stochastic gradient estimation and correlated update steps in the presence of graph data. This analysis contributes to the understanding of federated learning with graph data.

4. The authors conduct extensive experiments on multiple datasets and GNN backbones. The results demonstrate that GLASU achieves comparable performance to centralized models and reduce communication time.

**Weaknesses:**

1. The paper lacks statistical profiling results to precisely quantify the communication overhead caused by centralized training, despite the author's claim that it is significant.

2. Several concerns are raised regarding the problem setting of vertical federated learning as described in the paper:
    —The paper assumes that each client shares the same node set, but in real-world scenarios, it is more realistic to expect that nodes (e.g., users) from different departments only have a partial overlap. This contrasts with traditional federated learning where data from different clients can vary in sample numbers and distribution.
    —Privacy issue: While the authors briefly discuss the addition of differential privacy (DP) on node features, there is still a privacy concern with training labels, as the training labels need to be stored on all clients. Consider a scenario in which a department wishes to train a model using private node features from other departments. They would need to share their own private labels without any privacy guarantees.

3. GLASU employs lazy aggregation to reduce the number of synchronizations from $O(L)$ to $O(K)$. However, due to potential variations in edges across clients, this method does not address the straggler problem caused by unbalanced execution time before each synchronization.

4. Hyperparameter choice: The proposed lazy aggregation and staleness updates require the user to decide hyperparameters $K$ and $Q$. It's a question how to choose the right number as it may affect training accuracy. Moreover, how to select which layers to perform lazy aggregation is not discussed yet.

5. The experiment settings involve randomly sampling edges to form subgraphs for each client.  It would be beneficial for the authors to provide more clarification on the details of the splitting process and consider unbalanced splitting cases that resemble real-world scenarios.

6. Lack of experiment results: it's not very clear how the runtime is calculated. While the paper focuses on communication reduction, there is a lack of comparison in communication time between GLASU and baseline methods.

**Questions:**

— What is the practical scenarios when each client shares the same node set?

—How do you resolve the privacy concern raised in Point 2 of weaknesses?

—How to choose $K$ and $Q$ and how to select which layers to perform lazy aggregation?

—The convergence rate analysis is extended from Fedbcd and how does GLASU's convergence rate compare with Fedbcd?

---

### Official Review · Reviewer_771W · 2023-10-31

**Soundness:** 3 good
**Presentation:** 3 good
**Contribution:** 3 good
**Rating:** 5
**Confidence:** 3

**Summary:**

This paper addresses the communication challenges in GNN-structured federated learning. The authors propose two novel techniques aimed at minimizing communication overhead during model training. The first technique, termed "layer aggregation," involves utilizing local representations as inputs for subsequent layers, thereby diminishing the need for extensive communication between clients and the server. The second technique, named "stale updates," leverages outdated global information to update local models, further reducing communication costs. Additionally, the paper conducts theoretical convergence analysis and empirical experiments to validate the feasibility of these approaches.

**Strengths:**

Developing a framework capable of accommodating various GNN structures, with a specific focus on addressing communication reduction challenges, is a primary strength of this work. The authors supplement their framework by introducing two methods, thereby demonstrating the effectiveness of these techniques in terms of communication efficiency and model accuracy. Furthermore, the paper conducts a detailed analysis of the suggested algorithm, including a special case study of generality into other algorithm design.

**Weaknesses:**

Regarding the "stale updates" aspect, the authors employ a method in which they store the "all but m" representation by averaging the global information after subtracting this client's specific information. While this information extraction method is straightforward, it may not perform optimally in scenarios where embeddings are identical. In such cases, the model may struggle to discern which portion of the information was omitted.

**Questions:**

Is it sufficient to select only three clients for the experiment? Would the inclusion of a greater number of clients in the training process enhance the persuasiveness of the results?

---

### Official Review · Reviewer_XhfX · 2023-11-02

**Soundness:** 2 fair
**Presentation:** 3 good
**Contribution:** 2 fair
**Rating:** 3
**Confidence:** 4

**Summary:**

This paper studies vertical federated learning with graph data, where clients train a model based on partial features of the same set of samples they possess. The authors propose a model-splitting method and a communication-efficient algorithm, GLASU, which adopts lazy aggregation and stale updates to reduce communication overhead while ensuring convergence guarantees.

**Strengths:**

1.	This paper is generally clearly written.
2.	The proposed lazy aggregate-on and stale updates are simple and easy to implement.

**Weaknesses:**

1.The authors have not addressed the privacy issue, as the node embeddings and gradient information are transmitted directly. Although they mention that privacy protection mechanisms can be applied to the proposed GLASU algorithm, no experimental verification has been provided to assess the impact of such measures. This is a crucial aspect, considering that implementing a differential privacy model may lead to a decline in accuracy, while the use of homomorphic encryption could substantially reduce efficiency. It would be helpful if the authors could conduct an analysis or experiments to demonstrate the effect of incorporating privacy protection mechanisms.

2.	The proposed method heavily relies on the sampling method, as in every Q iterations, all the clients can only train the models based on the same set of sampled neighbors. Thus, it may not be easily adapted to GNNs with a global receptive field, such as graph transformers. Additionally, since the sampled set matters, different sampling strategies should be implemented to test the robustness of the stale updates.

3.	According to section 3.2, all layers in {l_1,…,l_K} are aggregated. However, in Algorithm 2, the routine ends when k=2 on the server, indicating that S_m [l_1] is not aggregated. This discrepancy needs clarification.

**Questions:**

1.	Could the authors explain in Table 2 why K=2 can achieve better accuracy than K=4?
2.	In the experiment settings, all the clients uniformly sample 80% of the edges from the whole graph, ensuring that each client has almost all the information of the graph. Would the proposed method still work under an unbalanced situation, such as when the number of edges follows a non-iid distribution?
3.	The test accuracy curve for GAT exhibits significant fluctuations. Does that imply that GLASU is sensitive to the GNN backbone or the neighborhood aggregation method?

---

### Meta-Review · Area_Chair_m67q · 2023-12-17

**Metareview:**

Summary:
This paper studies the intersection of two settings: training of graph neural networks, and vertical federated learning. The latter is a distributed training setting where the features of each sample are split across multiple client nodes, which have to cooperate to train a joint model. The former is a setting where sample are statistically related by a (given, known) graph.

The algorithm has somewhat limited novelty in the sense that it is starts with a standard GNN training algorithm, notices that it has a lot of communication overhead, and hence makes communication lazy.

Strengths: clear writing

Weaknesses:
lack of clear motivation for the problem
limited novelty in the algorithm
lack of comparison to any baselines

**Justification For Why Not Higher Score:**

Low reviewer scores, and lack of any rebuttal to address the reviewer concerns.
Lack of novelty in the algorithm
Lack of clear motivation for the problem considered

**Justification For Why Not Lower Score:**

is already lowest score

---

### Decision · Program_Chairs · 2024-01-16

Reject